# Cryo-EM structures of the human band 3 transporter indicate a transport mechanism involving the coupled movement of chloride and bicarbonate ions

Chih-Chia Su[1]\*, Zhemin Zhang[1], Meinan Lyu[1], Meng Cui[2], Edward W. Yu[1]\*

**1** Department of Pharmacology, Case Western Reserve University School of Medicine, Cleveland, Ohio, United States of America, **2** Department of Pharmaceutical Sciences, Northeastern University School of Pharmacy, Boston, Massachusetts, United States of America

\* cxs670@case.edu (C-CS); edward.w.yu@case.edu (EWY)

**Data Availability Statement:** Atomic coordinates and structure factors have been deposited with accession codes 8T3R (PDB, https://www.rcsb.org/structure/8T3R) and EMD-41012 (EMBD,

## Abstract

The band 3 transporter is a critical integral membrane protein of the red blood cell (RBC), as it is responsible for catalyzing the exchange of bicarbonate and chloride anions across the plasma membrane. To elucidate the structural mechanism of the band 3 transporter, detergent solubilized human ghost membrane reconstituted in nanodiscs was applied to a cryo-EM holey carbon grid to define its composition. With this approach, we identified and determined structural information of the human band 3 transporter. Here, we present 5 different cryo-EM structures of the transmembrane domain of dimeric band 3, either alone or bound with chloride or bicarbonate. Interestingly, we observed that human band 3 can form both symmetric and asymmetric dimers with a different combination of outward-facing (OF) and inward-facing (IF) states. These structures also allow us to obtain the first model of a human band 3 molecule at the IF conformation. Based on the structural data of these dimers, we propose a model of ion transport that is in favor of the elevator-type mechanism.

## Introduction

The band 3 anion transporter, also known as anion exchanger 1 (AE1), belongs to the solute carrier 4A family of membrane proteins [1]. In human, the gene *SLC4A1* encodes the band 3 protein which is responsible for catalyzing the electroneutral process of bicarbonate and chloride exchange across the plasma membrane with a stoichiometry of 1:1 bicarbonate-to-chloride molar ratio [2,3]. Band 3 is a major membrane protein of the red blood cell (RBC), with each red cell containing approximately 1 million copies of this membrane protein [4]. Band 3 appears to have 2 main functional roles; it facilitates carbon dioxide transport via anion exchange between bicarbonate and chloride, and maintains the shape of the erythrocyte by serving as a mediator for protein–protein interactions that couple the lipid bilayer to the underlying membrane skeleton [3,5,6]. In addition, band 3 is highly expressed in the basolateral face of α-intercalated cells of the collecting dusts of nephrons in the kidney. Interestingly,

https://www.emdataresource.org/EMD-41009) for IF-OF-Cl-; 8T3U (PDB, https://www.rcsb.org/structure/8T3U) and EMD-41012 (EMDB, https://www.emdataresource.org/EMD-41012) for IF-IF-Cl-; 8T44 (PDB, https://www.rcsb.org/structure/8T44) and EMD-41019 (EMDB, https://www.ebi.ac.uk/emdb/EMD-41019) for IF-OF-HCO$_3^-$; 8T45 (PDB, https://www.rcsb.org/structure/8T45) and EMD-41020 (EMDB, https://www.ebi.ac.uk/emdb/EMD-41020) for IF-IF-HCO$_3^-$; 8T47(PDB, https://www.rcsb.org/structure/8T47) and EMD-41023 (EMDB, https://www.ebi.ac.uk/emdb/EMD-41023) for OF-OF-HCO$_3^-$.

**Funding:** This work was supported by the National Institutes of Health (R01AI145069 to EWY). The funders had no role in study design, data collection and analysis, decision to publish, or preparation of the manuscript.

**Competing interests:** The authors have declared that no competing interests exist.

**Abbreviations:** AE1, anion exchanger 1; BaR, build and retrieve; CDS, correlated double sampling; CHS, cholesteryl hemisuccinate; CTF, contrast transfer function; GBIM, generalized Born with implicit membrane; IF, inward-facing; MD, molecular dynamics; OF, outward-facing; PBS, phosphate-buffered saline; PC, phosphatidylcholine; PME, particle mesh Ewald; RBC, red blood cell; RMSD, root mean square deviation.

this isoform appears to lack the first 65 N-terminal amino acids [7]. In this organ, it helps regulate intracellular pH, cell volume, and membrane potential [1]. Loss of function of band 3 often results in severe hemolytic anemia and premature death [8,9]. Mutations of the band 3 protein are also linked to particular cases of hereditary spherocytosis [10] and hereditary stomatocytosis [9], and all cases of Southeast Asian ovalocytosis [11,12].

Human erythrocyte band 3 consists of 911 amino acids that form a 110 kDa glycoprotein. Band 3 can be divided into 2 domains: the N-terminal cytoplasmic (residues 1–360) and C-terminal transmembrane (residues 361–911) domains [13]. The cytoplasmic domain of band 3 is promiscuous. It has been shown that this domain can interact with a number of cytoplasmic proteins of the RBC, including the ankyrin-1-spectrin and actin junctional complex [14,15]. In addition, the transmembrane domain of this transporter plays an important role in exporting bicarbonate out of the cell by exchanging this ion with chloride for charge neutralization [16]. These 2 domains are connected by a linker region containing a trypsin-sensitive site at residue K360 that can be readily cleaved by mild protease treatment [17,18]. Interestingly, it appears that the membrane domain (residues 361–911) alone without the cytoplasmic domain (residues 1–360) is fully functional in transporting substrates [19]. Structures of both the cytoplasmic domain [20,21] and transmembrane domain [22] of band 3 have been separately determined using X-ray crystallography. Recently, a cryo-EM structure of the erythrocyte ankyrin-1 complex purified from human erythrocytes has been reported [23], suggesting that three band 3 dimers are involved in forming this complex. In addition, a cryo-EM structure of bovine band 3 has been resolved and captured multiple conformational states of this membrane protein [24].

To elucidate the structural mechanism of the band 3 transporter, we used the ghost membrane of the human RBC, where band 3 is predominantly embedded in this membrane and accounts for approximately 30% of the protein component [4]. We recently developed a cryo-EM methodology named "Build and Retrieve" (BaR) [25]. This methodology is capable of simultaneously identifying and solving near-atomic resolution structures of various membrane proteins from crude cell membrane. BaR is an iterative methodology capable of performing in silico purification and sorting of images of several different classes of biomacromolecules within a large heterogeneous data set. It is powerful in that it not only allows for deconvoluting images of a mixture of proteins, but it can also simultaneously produce near atomic resolution cryo-EM maps for an individual protein in various conformational states, all from a heterogeneous, multiprotein system. Using this approach, we solved several cryo-EM structures of the band 3 transporter. We also obtained the first structural feature of human band 3 at the inward-facing conformational state, although the inward-facing structure of bovine band 3 has been reported recently [24]. We here present 5 structures of dimeric band 3, either alone or bound with Cl- or HCO$_3^-$. We observed that band 3 can form both symmetric and asymmetric dimers. On the basis of our findings, we propose a mechanistic model of ion transport that is favorable to the elevator-type mechanism.

## Results

### Cryo-EM structures of the band 3 transporter in the presence of Cl-

To help elucidate the structures of the band 3 membrane protein in a more native environment, we employed the ghost membrane of the human RBC solubilized in detergent micelles. We reconstituted the detergent solubilized raw membrane sample into nanodiscs and enriched this protein-nanodisc sample by size exclusion chromatography using a buffered solution containing 100 mM Cl- ion. From this sample, we obtained a major peak with sizes corresponding

to 300–600 kDa. We then collected single-particle cryo-EM images of this enriched peak and processed the cryo-EM data using the BaR methodology.

Surprisingly, based upon the cryo-EM images, we only observed 1 single species of protein-nanodisc particles of high abundance in this cryo-EM sample. The BaR protocol allowed us to identify that these particles correspond to the C-terminal transmembrane domain of the band 3 transporter (Fig 1 and S1 Table). Further classification of the single-particle cryo-EM images revealed that there were 2 distinct populations with different conformations in the nanodisc sample (S1 Fig). Interestingly, one of these structures depicted that the band 3 transporter assembles as an asymmetric dimer with a very distinct conformational state in each subunit, where one of the subunits displays an outward-facing (OF) conformation and the other

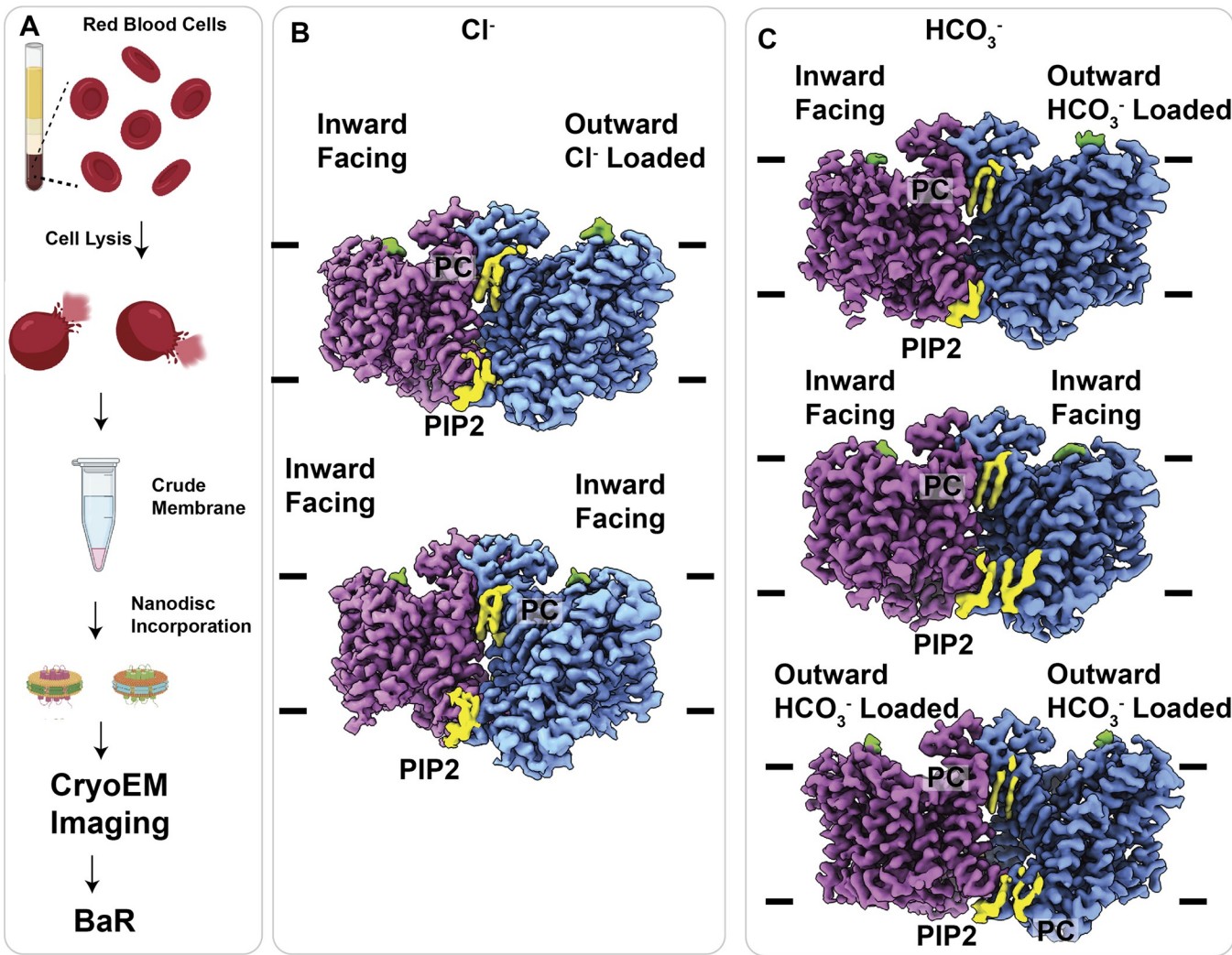

**Fig 1. Structural determination of band 3. Cryo-EM maps of band 3.** (A) The BaR protocol for studying human RBC. The raw ghost membrane of the human RBC was solubilized in detergent micelles and reconstituted into nanodiscs. The protein-nanodisc sample was enriched size exclusion chromatography using a buffered solution containing 100 mM Cl⁻ or HCO₃⁻. Single-particle cryo-EM images were collected, and the cryo-EM data were processed using the BaR methodology. (B) Cryo-EM density maps of the band 3 in the presence of Cl⁻. The BaR protocol led us to solve 2 structures of band 3 at the OF-IF and IF-IF conformational states. The resolutions of these 2 structures are 2.99 Å and 2.97 Å, respectively. (C) Cryo-EM density maps of the band 3 in the presence of HCO₃⁻. The BaR protocol led us to solve 3 structures of band 3 at the IF-IF, OF-IF, and OF-OF conformational states. The resolutions of these 2 structures are 2.99 Å, 3.12 Å, and 3.16 Å, respectively. The clip arts of Fig 1A are provided by BioRender (https://www.biorender.com). BaR, build and retrieve; IF, inward-facing; OF, outward-facing; RBC, red blood cell.

subunit exhibits an inward-facing (IF) state. The other band 3 structure in the sample depicts a symmetric dimer with each subunit presenting an IF conformation (S1 Fig). These 2 structures were assigned as an asymmetric OF-IF dimer and a symmetric IF-IF dimer. We then refined cryo-EM structures of these 2 dimers. The final model of each subunit contains all of the C-terminal transmembrane domain residues (residues 372–888), except for residues 743–751 because of weak corresponding cryo-EM densities.

**Structure of OF-IF dimeric band 3.** The transmembrane domain of band 3 is dimeric in form with both the N- and C-termini pointing toward the inside of the RBC. Each subunit (A and B) contains 6 amphipathic α-helices (H1-H6 and H1′-H6′, respectively) running in parallel with the membrane plane and 14 transmembrane helices (TM1-TM14 and TM1′-TM14′, respectively) spanning the ghost membrane (Fig 2A). TMs 1–7 and TMs 8–14 are structurally related to each other in that they are arranged into 2 inverted repeats (S2 Fig). Of the 14 TM helices, TM3 and TM10 are relatively short and can only span a portion of the ghost membrane. These 2 TMs are structurally arranged in such a way that they vertically stack against each other, leaving the gap between these 2 TMs to form a crossover region with an internal cavity within the membrane. The TMs and Hs of the C-terminal domain of band 3 are

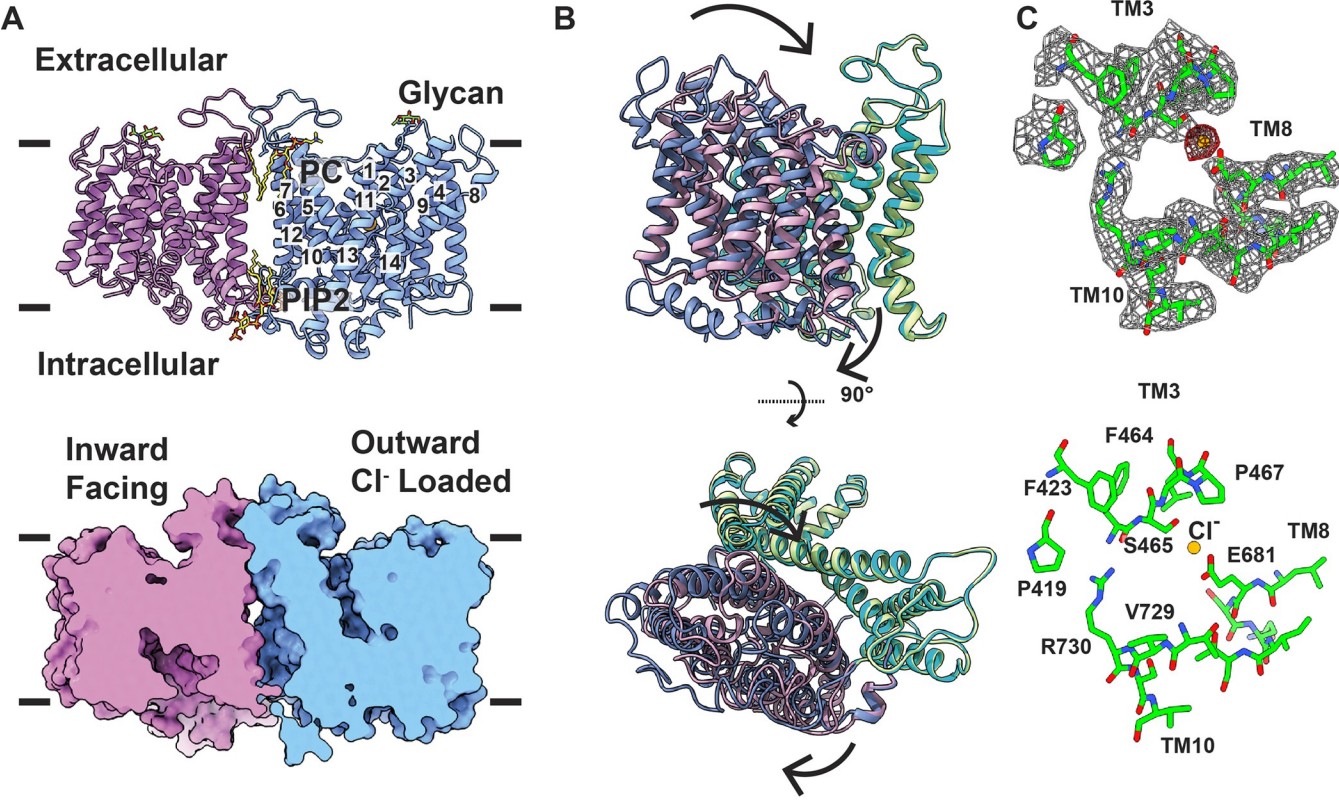

**Fig 2. Structure of the OF-IF state of band 3 in the presence of Cl⁻.** (A) Side view of the ribbon diagram of the band 3 dimer viewed in the membrane plane (upper panel). Subunits A and B of band 3 are colored cyan and pink. The bound lipid molecules (PC and PIP2) located at the subunit interface are in sticks. The NAG moiety is highlighted with splitpea colored sticks. The bound Cl⁻ in subunit A is shown in orange sphere. The surface representation of the band 3 dimer is included (lower panel). This figure reveals that the structure of subunit A (cyan) is captured in an OF conformation, whereas the structure of subunit B (pink) is captured in an IF conformation. (B) Superimposition of the structures of subunits A and B. This superimposition indicates that the interface domain of subunit A (green) and subunit B (blue) can be easily superimposed. However, the transport domain (pink, subunit A; cyan, subunit B) seemingly performs a rigid body rocking motion to switch the conformation of the band 3 subunit between the OF and IF states. (C) Cryo-EM densities of the Cl⁻ binding site. The Cl⁻ binding site is located at the crossover region formed by TM3 and TM10. Bound Cl⁻ (yellow) is 3.3 Å away from the side chain oxygen of E684. Cryo-EM densities of bound Cl⁻ and local residues are in red and gray meshes. Residues surrounding the Cl⁻ binding site are highlighted with green sticks. IF, inward-facing; OF, outward-facing.

designated numerically from the N- to C-termini: H1 (380–400), TM1 (403–430), TM2 (437–454), TM3 (467–482), TM4 (486–507), H2 (509–516), TM5 (518–546), TM6 (570–593), TM7 (599–623), TM8 (644–689), TM9 (702–719), TM10 (728–738), TM11 (761–773), H3 (777–782), TM12 (785–800), H4 (804–813), H5 (823–827), TM13 (830–852), TM14 (856–877), and H6 (880–886).

The dimer interface of band 3 is primarily composed of TM5, TM6, the loop region between TM5 and TM6, and the horizontal helix H4 and loop between TM12 and TM13, as well as their counterpart elements TM5′, TM6′, TM12′, TM13′, H4′, and their corresponding flexible loops. Interestingly, 2 extra densities are observed at the interface between the 2 subunits of the band 3 dimer. These extra densities are located nearby the extracellular surface and buried within the membrane. The shape of each extra density fits well with a phosphatidylcholine (PC) lipid. We therefore used 2 PC molecules to model these 2 extra densities (Figs 2A and S3). Likewise, there are another 2 extra densities located at the junction of the 2 band 3 subunits and close to the inner surface of the ghost membrane. These 2 densities resemble 2 individual phosphatidylino-sitol 4,5-bisphosphate (PIP2) lipids and we include 2 PIP2 molecules at the interface of the 2 band 3 subunits (Figs 2A and S3). Posttranslational modifications have been found to play a role in the stability of many eukaryotic proteins. Within the structure of band 3, an N-linked glycosylation site is found at residue N642 of each subunit of this transporter. It is observed that each asparagine is connected to an N-acetylglucosamine moiety (NAG) (Figs 2A and S3).

Although band 3 displays as a dimer, there is strong evidence that the 2 subunits within the dimer function independently as transport studies indicate that blocking of one subunit of the transporter does not prevent the shuttling of substrates by the other subunit [1,26]. Like the crystal structure of the C-terminal domain of band 3 [22], our cryo-EM structure indicates that the transmembrane region of this transporter can be separated into a transport domain and an interface domain. The transport domain consists of TMs 1–4 and their inverted repeat TMs 8–11, where these TMs are tightly bundled together. The interface domain is made up of TMs 5–7 and their inverted repeat TMs 12–14, and these TMs are responsible for creating a subunit interface of the band 3 dimer. Transport of anions across the membrane is presumed to occur at the interface between the transport and interface domains [1,22]. Interestingly, we observed that a long flexible loop containing 40 amino acids (residues 624–663) is engaged in connecting the transport and interface domains. This long flexible loop may be important for the functional dynamics of this transporter.

Surprisingly, our structure indicates that each subunit of the band 3 dimer displays a very distinct conformational state, leading to the formation of an asymmetric dimer. Superimposi-tion of subunits A and B within the dimer gives rise to a large r.m.s.d. of 2.0 Å, suggesting that there is a significant structural difference between the two (Fig 2B). It appears that subunit A of band 3 exhibits an OF conformation, whereas subunit B of this transporter depicts an IF state. A detailed comparison of the structure of these 2 subunits reveals that the interface domain of the 2 subunits can be easily superimposed. However, the relative location of the transport domain of these two subunits in relation to the interface domain is very different. This shift in location can be interpreted as a rigid-body movement of the transport domain with respect to the interface domain of the transporter when compared with the structures of these 2 different conformational states. This rigid-body motion may be accounted for shifting the entire transport domain and switching the conformation of the transporter between the OF and IF states (Fig 2B and S1 Movie). The long flexible loop (residues 624–663), which links TM7 and TM8, is ideal to provide the flexibility for this interconversion.

Recently, a cryo-EM structure of the bovine band 3 dimer, which also includes OF and IF subunits, has been determined. It appears that our human band 3 structure is in good agree-ment with that of bovine band 3. Superimposition of the structure of the OF subunit of human

band 3 to that of bovine band 3 gives rise to an r.m.s.d. of 1.07 Å (for 422 Cα atoms). Likewise, superimposition of the structure of the IF subunit of human band 3 to that of bovine band 3 provides an r.m.s.d. of 1.28 (for 422 Cα atoms) (S4 Fig).

An extra density is observed within the interface between the transport and interface domains of subunit A of band 3. This subunit displays an OF conformational state. The extra density is spherical in shape and is compatible with a bound ion (Fig 2C). Apparently, this extra density is found to directly interact with residue E681 of subunit A of band 3. There is strong evidence that residue E681 critically participates in the anion exchange pathway [27,28]. Interestingly, this density is absent in the bicarbonate sample without containing any Cl⁻ ions (see below). We therefore assigned this extra density as a bound Cl⁻ ion. The bound Cl⁻ ion is 3.3 Å away from the carboxylate oxygen of the side chain of E681. The binding of this ion is anchored by at least 8 residues, including F423, F464, P419, S465, P467, E681, V729, and R730, which provide hydrophobic and electrostatic interactions to secure Cl⁻ binding (Fig 2C). A channel-like cavity is formed in subunit A of band 3, allowing the bound Cl⁻ ion to become accessible to the extracellular environment (Fig 2A). No extra cryo-EM density was found in subunit B of band 3, where this subunit displays an IF conformation. This suggests that the IF conformational subunit does not contain any bound ion or ligand.

It is a little surprising that our cryo-EM structure indicates that Cl⁻ ion is anchored by an acidic glutamate residue. However, this is actually not uncommon to find the involvement of negatively charged aspartate or glutamate residue to participate in Cl⁻ binding. For example, a model of the Cl⁻ binding site in the *Aedes aegypti* AAT1 transporter is composed of an aspartate residue [29]. Indeed, it has been observed that there are numerous halide ion binding sites, including those for Cl⁻, Br⁻, and I⁻, each of which contains a negatively charged aspartate or glutamate residue that is actively engaged in halide binding [30]. It is also interesting to note that the ion conductance pathway of CLC channels, such as ClC-1, contain a glutamate-tyrosine site in the midway of the channel. It has been proposed that chloride conductance may involve the shuttling of Cl⁻ to the glutamate-tyrosine site [31].

To validate this chloride-binding site in the band 3 exchanger, we performed 1,000 ns molecular dynamics (MD) simulations (S5 Fig) based on the Cl⁻ binding site of our cryo-EM structure. In agreement with our Cl⁻ bound band 3 structure, MD simulations indeed indicate that Cl⁻ is anchored stably in this ion-binding site with strong interactions with a number of band 3 residues surrounding it (S2 Table). Our MD simulations also suggest that this bound Cl⁻ ion is coordinated with 6 water molecules to further stabilize the binding (S6 Fig).

**Structure of IF-IF dimeric band 3.** In this structure, the 2 band 3 subunits within the dimer are identical in conformation, displaying the IF conformational state of the membrane protein (Figs 3A and S7). A channel-like feature is formed in each band 3 subunit, where this channel spans the inner leaflet of the ghost membrane and connects the intracellular space up to the mid-point of the interior of the transmembrane region of this transporter (Fig 3A). No extra cryo-EM densities were found within the channel of each band 3 subunit, suggesting that there are no bound ions at this conformational state.

## Structures of band 3 in the presence of HCO₃⁻

We also determined structures of band 3 in a buffered solution containing 100 mM HCO₃⁻ ion. The BaR methodology allowed us to distinguish 3 different dimeric structures of this transporter. These 3 structures were classified as IF-IF, OF-IF, and OF-OF dimers (S8 Fig and S1 Table).

**Structure of IF-IF dimeric band 3.** The structure of IF-IF band 3 depicts that the 2 band 3 subunits present an IF conformational state (Figs 3B and S9). An IF channel is present in each subunit. This channel connects the interior space of the red cell to the mid-point of the

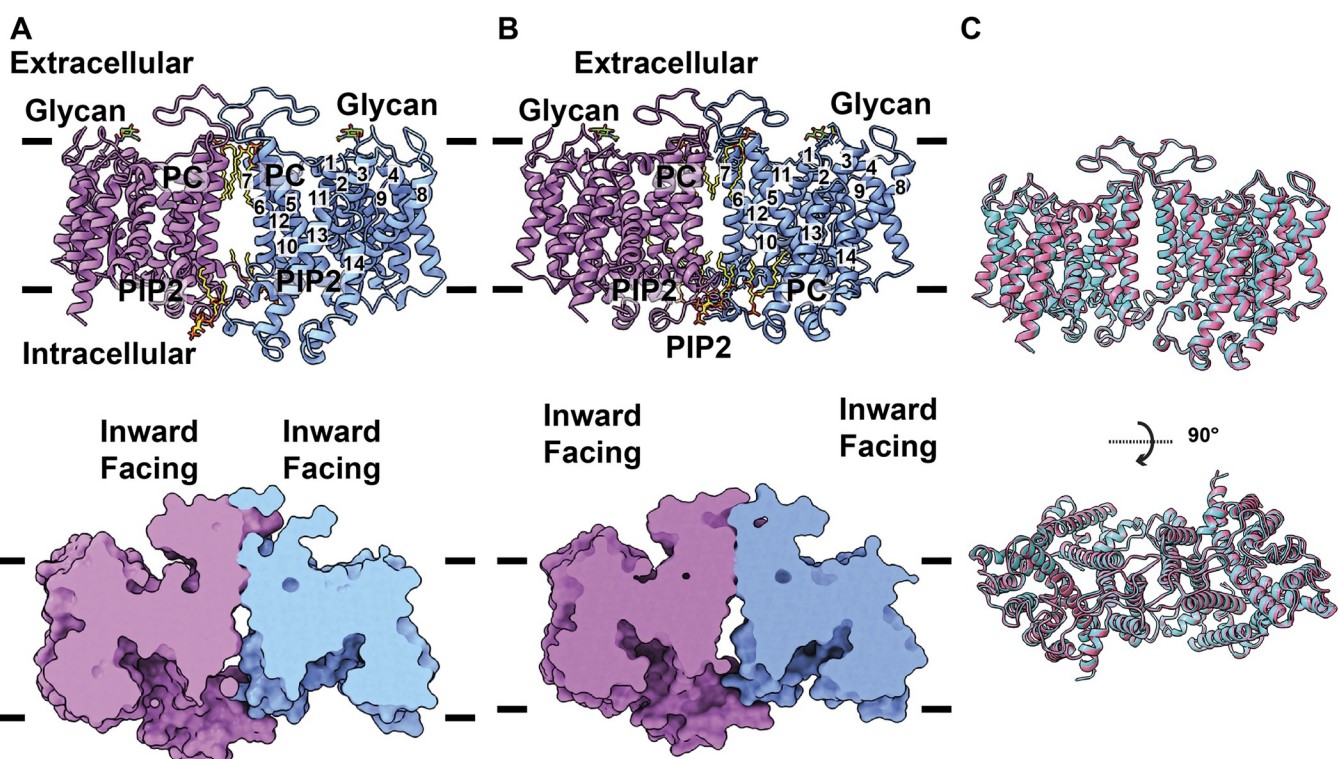

**Fig 3. Structure of the IF-IF state of band 3 in the presence of Cl⁻ or HCO₃⁻.** (A) Side view of the ribbon diagram of the band 3 dimer (in the presence of 100 mM Cl⁻) viewed in the membrane plane (upper panel). Subunits A and B of band 3 are colored cyan and pink. The 4 bound lipid molecules (PC and PIP2) located at the subunit interface are in sticks. The NAG moiety is highlighted with splitpea colored sticks. The surface representation of the band 3 dimer is included (lower panel). This figure reveals that both the structures of subunit A (cyan) and subunit B (pink) are captured in an IF conformation. (B) Side view of the ribbon diagram of the band 3 dimer (in the presence of 100 mM HCO₃⁻) viewed in the membrane plane (upper panel). Subunits A and B of band 3 are colored cyan and pink. The bound lipid molecules (PC and PIP2) located at the subunit interface are in sticks. The NAG moiety is highlighted with splitpea colored sticks. The surface representation of the band 3 dimer is included (lower panel). This figure reveals that both the structures of subunit A (cyan) and subunit B (pink) are captured in an IF conformation. (C) Superimposition of the structures of the 2 IF-IF band 3 dimers. The IF-IF dimer in the presence of Cl⁻ and IF-IF dimer in the presence of HCO₃⁻ are colored cyan and pink, respectively. The superimposition indicates that the conformations of these 2 dimers are nearly identical to each other. IF, inward-facing.

transmembrane domain of band 3 and allows for the interior of the transmembrane domain to be exposed to solvent (Fig 3B). No extra cryo-EM density is observed within each subunit of the band 3 dimer, suggesting that no bound ion or ligand is present in the transporter. This IF-IF dimeric structure is very similar to that of the IF-IF dimer in the presence of 100 mM Cl⁻. Superimposition of these 2 dimers give rise to an r.m.s.d. of 0.6 (Fig 3C), suggesting that these 2 structures depict an identical transient state of the transporter.

**Structure of OF-IF dimeric band 3.** The overall conformation of this band 3 structure is similar to the OF-IF, Cl⁻ bound structure (Figs 4A, 4B, and S10). Superimposition of these 2 dimers results in an r.m.s.d. of 0.6 Å, indicating that these 2 OF-IF band 3 dimers represent very similar transient conformational states (Fig 4C). A detailed inspection reveals that an extra density is found within the OF channel formed by subunit A of dimeric band 3. The shape of this extra density is compatible with an HCO₃⁻ ion (Fig 4B). Interestingly, this extra density is absent in the 2 band 3 structures determined in the presence of Cl⁻ only. It is also worth noting that the spherical extra density representing the bound Cl⁻ ion in the chloride loaded OF-IF structure is absence in this dimeric band 3 structure. The side chain nitrogen of R730 is 3.0 Å away from bound HCO₃⁻, contributing to form a hydrogen bond with this ligand. In addition, the backbone nitrogen of V729 and bound HCO₃⁻ are 2.7 Å apart, creating

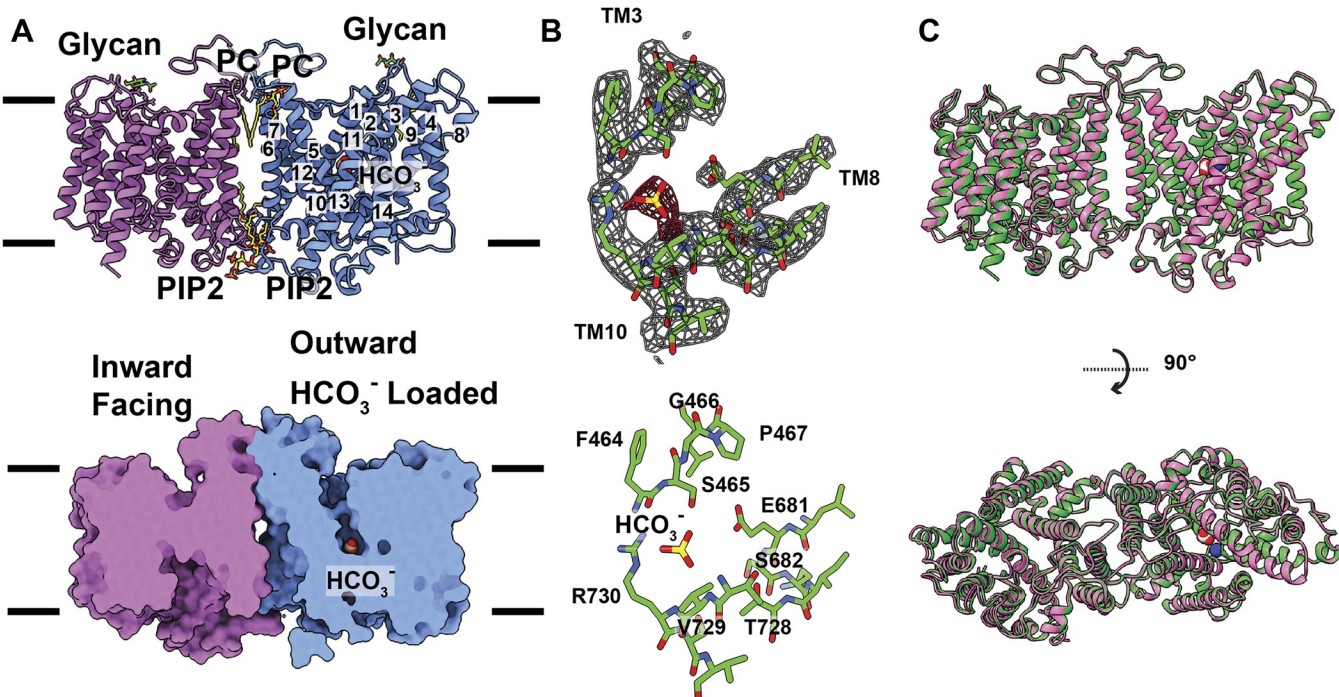

**Fig 4. Structure of the OF-IF state of band 3 in the presence of HCO$_3^-$.** (A) Side view of the ribbon diagram of the band 3 dimer viewed in the membrane plane (upper panel). Subunits A and B of band 3 are colored cyan and pink. The bound lipid molecules (PC and PIP2) located at the subunit interface are in sticks. The NAG moiety is highlighted with splitpea colored sticks. The bound HCO$_3^-$ in subunit A is shown in spherical models (yellow, carbon; red, oxygen). The surface representation of the band 3 dimer is included (lower panel). This figure reveals that the structure of subunit A (cyan) is captured in an OF conformation, whereas the structure of subunit B (pink) is captured in an IF conformation. (B) Cryo-EM densities of the HCO$_3^-$ binding site. The HCO$_3^-$ binding site is located at the crossover region formed by TM3 and TM10. Bound HCO$_3^-$ (yellow, carbon; red, oxygen) is 3.0 Å away from the sidechain nitrogen of R730 and 2.7 Å away from the backbone nitrogen of V729. Cryo-EM densities of bound HCO$_3^-$ and local residues are in red and gray meshes. Residues surrounding the HCO$_3^-$ binding site are highlighted with green sticks. (C) Superimposition of the structures of the OF-IF band 3 dimer in the presence of HCO$_3^-$ (green) and the OF-IF band 3 dimer in the presence of Cl$^-$ (pink). The superimposition indicates that the conformations of these 2 dimers are nearly identical to each other. IF, inward-facing; OF, outward-facing.

an α-helix dipolar interaction [32] to secure the binding. Residues involved in forming this HCO$_3^-$ include F464, S465, G466, P467, E681, S682, T728, V729, and R730, which mainly provide hydrophobic and electrostatic interactions to stabilize HCO$_3^-$ binding. Interestingly, no extra density was found within the IF channel of the subunit B of band 3, suggesting that there is no ion bound in this channel.

**Structure of OF-OF dimeric band 3.** This OF-OF structure presents a symmetric dimer of band 3, where the structures of the 2 subunits are identical to each other (Figs 5A and S11). Each band 3 subunit carries an OF channel that connects the interior of the top half of the transmembrane region of this membrane protein to the extracellular space. An extra density exhibiting the presence of a bound HCO$_3^-$ ion is found in each channel (Fig 5B). This HCO$_3^-$ binding site is similar to but also distinct from that found in subunit A of the OF-IF structure. The sidechain nitrogen of R730 and sidechain oxygen of E681 are 3.9 Å and 3.3 Å away from bound HCO$_3^-$ (Fig 5B). Residues participating in this HCO$_3^-$ binding include F464, S465, G466, P467, E681, S682, T728, V729, and R730. In comparison with bound HCO$_3^-$ in the OF-IF structure, this bound HCO$_3^-$ ion (from each subunit of the OF-OF structure) is found to shift its location toward E681 by 2.1 Å and partially occupy both the HCO$_3^-$ binding site and Cl$^-$ binding site observed from the OF-IF structures of band 3 in the presence of HCO$_3^-$ and Cl$^-$, respectively. This may imply that the binding of HCO$_3^-$ and Cl$^-$ are mutually exclusive in the same subunit of band 3.

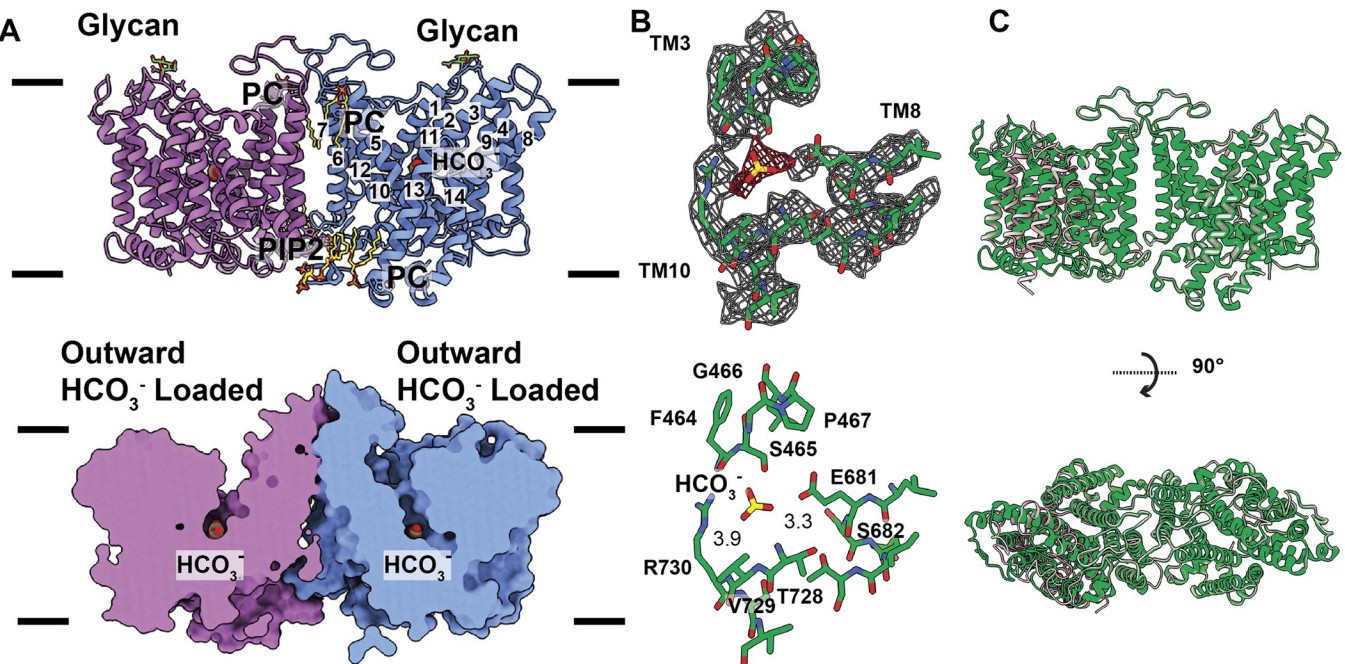

**Fig 5. Structure of the OF-OF state of band 3 in the presence of HCO$_3^-$.** (A) Side view of the ribbon diagram of the band 3 dimer viewed in the membrane plane (upper panel). Subunits A and B of band 3 are colored cyan and pink. The bound lipid molecules (PC and PIP2) located at the subunit interface are in sticks. The NAG moiety is highlighted with splitpea colored sticks. The bound HCO$_3^-$ ions in both subunits A and B are shown in spherical models (yellow, carbon; red, oxygen). The surface representation of the band 3 dimer is included (lower panel). This figure reveals that the structures of both subunit A (cyan) and subunit B (pink) are captured in an OF conformation. The bound HCO$_3^-$ ions are in spherical models. (B) Cryo-EM densities of the HCO$_3^-$ binding site in subunit A of band 3. The HCO$_3^-$ binding site is located at the crossover region formed by TM3 and TM10. Bound HCO$_3^-$ (yellow, carbon; red, oxygen) is 3.3 Å from the sidechain oxygen of E681 and 3.9 Å away from the sidechain nitrogen of R730. Cryo-EM densities of bound HCO$_3^-$ and local residues are in red and gray meshes. Residues surrounding the HCO$_3^-$ binding site are highlighted with green sticks. The HCO$_3^-$ binding site in subunit B of band 3 is identical to that of subunit A. (C) Superimposition of the structures of the OF-OF band 3 dimer in the presence of HCO$_3^-$ (green) and the OF-IF band 3 dimer in the presence of HCO$_3^-$ (pink). The superimposition indicates that the conformations of the subunit A of these 2 dimers are nearly identical to each other. However, the conformations of the subunit B of these 2 dimers are quite distinct from each other. IF, inward-facing; OF, outward-facing.

## Discussion

In this study, we have defined cryo-EM structures of the band 3 exchanger in the presence of Cl$^-$ or HCO$_3^-$. Based on the structural information, we identified both the Cl$^-$ and HCO$_3^-$ binding sites. These 2 binding sites are distinct from each other. However, they are in close proximity, approximately 5.2 Å apart. These 2 binding sites may communicate with each other via the bound Cl$^-$ and HCO$_3^-$ ions. It appears that the binding of Cl$^-$ is mainly governed by the interaction between the charged residue E681 and bound Cl$^-$. Likewise, it is observed that the charged residue R730 is responsible for anchoring the bound HCO$_3^-$ ion. Interestingly, there are several polar residues, including S465, T727, and S731, located at the interface between the Cl$^-$ and HCO$_3^-$ binding sites. It is possible that these polar residues may maintain and secure the communication between these 2 sites by switching the orientation of their hydroxyl side chains.

Surprisingly, we only observed these bound ions in the OF conformation, and not in the IF state, of the band 3 protomers (see below). In comparison with the OF and IF conformations, it appears that there are significant changes in terms of structure and relative position of TM3 and TM10. As mentioned, the gap between the 2 short TMs, TM3 and TM10, form a crossover region within the membrane. In the OF conformation, the bound Cl$^-$ ion is found to house in this crossover region. In view of the IF conformational state, the relative position of the entire

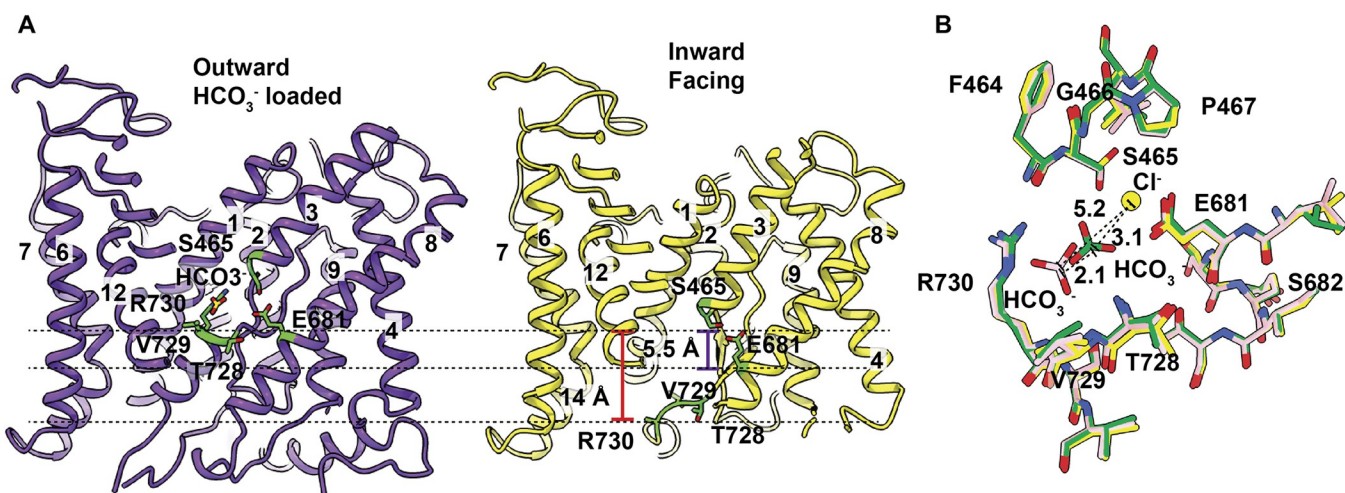

**Fig 6. Comparison of the OF and IF conformations and ion-binding sites of the HCO$_3^-$ and Cl$^-$ ions.** (A) Comparison of the OF and IF conformations of the band 3 subunit. At the IF state, residues E681 and R730 are found to shift toward the interior of the cell by 5.5 Å and 14 Å when compared with those in the OF conformation. (B) Superimposition of the ion binding sites from the structures of IF-OF-HCO$_3^-$, OF-OF-HCO$_3^-$, and IF-OF-Cl$^-$ dimers. This superimposition suggests that the position of the bound HCO$_3^-$ ion in the OF-OF-HCO$_3^-$ structure is located between the positions of bound HCO$_3^-$ in the IF-OF-HCO$_3^-$ structure and bound Cl$^-$ in the IF-OF-Cl$^-$ structure. As residue E681 is found to utilize for both HCO$_3^-$ and Cl$^-$ binding, this superimposition may imply that the binding of HCO$_3^-$ and Cl$^-$ are mutually exclusive from each other (IF-OF-HCO$_3^-$, IF-OF conformation of band 3 in the presence of HCO$_3^-$; OF-OF-HCO$_3^-$, OF-OF conformation of band 3 in the presence of HCO$_3^-$; IF-OF-Cl$^-$, IF-OF conformation of band 3 in the presence of Cl$^-$). IF, inward-facing; OF, outward-facing.

TM3 seems to shift toward the inside of the cell by 5 Å in relation to the OF conformation, resulting in closing up this crossover region. It is observed that there is also a drastic conformational change in TM10 in accompanying with the shift of TM3. Surprisingly, residues forming TM10 display as a random loop in the IF conformation. This loop is quite random and disordered, leaving residues 735–753 untraceable in our cryo-EM structure. At this state, residue R730 drastically shifts in position, moving toward the inside of the cell and exposing itself to solvent. The net result is that the HCO$_3^-$ binding site, formed by R730 and V729, is observed to be dissembled at this state. Based on the structural information of the IF subunit, both E681 and R730 are observed to shift toward the inside of the RBC by approximately 5.5 Å and 14 Å, respectively, compared with the conformation of the OF subunit. It appears that both Cl$^-$ and HCO$_3^-$ ions have been released from their corresponding binding sites at this conformational state (Fig 6). Because of the closing of the internal cavity at the crossover region and dissembling of ion binding sites at the IF conformational state, the binding of these ions at this state may be more transient and lower in affinity. Based on the structural information, the ions are possibly more accessible to solvent at this state. Therefore, it may be difficult to capture the bound Cl$^-$ or HCO$_3^-$ at the IF state.

It is likely that our cryo-EM structures capture different transient states of the transporter within the transport cycle. In comparison with the structures of the IF and OF band 3 protomers, it is possible that the transport domain is capable of performing a rigid-body rocking movement during Cl$^-$ and HCO$_3^-$ transport. The concentration of Cl$^-$ is higher outside the RBC. However, the concentration of HCO$_3^-$ should be lower on the outside compared with that of the inside of the cell. At the OF conformational state, the charged residues, including E535, K539, and K542 of the interface domain, and E429, K430, and E473 of the transport domain, are exposed to solvent and nearby the opening of the OF channel. These charged residues have been proposed to form salt bridges [33] to stabilize different conformational states of the transporter. Based on the structural information, these charged residues may also be

critical for ion transport by facilitating the shuttle of $Cl^-$ into and $HCO_3^-$ out of the OF channel.

The first step for $Cl^-$ transport may involve the accessibility of $Cl^-$ into the transporter via the OF state (Fig 7). $Cl^-$ could then bind strongly in the $Cl^-$ binding site formed by residues E681 and G466. Binding of $Cl^-$ may trigger a conformational change and switch the band 3 transporter to the IF state. This would occur via a rigid-body rocking movement of the transport domain with respect to the channel domain of band 3. This movement would also lead to a downward shift of the $Cl^-$ binding site and the bound $Cl^-$ ion by approximately 5.5 Å toward the inside of the RBC (Fig 6). The bound $Cl^-$ ion could then be released into the cell (Fig 7). In this IF state, residue R730 is located at the surface of the inner leaflet of the ghost membrane. This position allows R730 to be accessible to solvent and help recruit $HCO_3^-$ from the inside of the cell. The deliverance of $Cl^-$ into and collection of $HCO_3^-$ from the RBC via the IF conformational state of band 3 could be coupled with each other (Fig 7), because charge-charge repulsion between these 2 ions could help expel $Cl^-$ into the cell. After releasing $Cl^-$ and picking up $HCO_3^-$, the band 3 transporter may then shift its conformation back to the OF state. This movement is also caused by a rigid-body rocking motion of the transport domain in relation to the channel domain (S1 Movie), resulting in an upward shift of the location of R730 by approximately 14 Å toward the outside of the RBC (Figs 6 and 7). In this conformational state, the bound $HCO_3^-$ ion is anchored by residues R730 and V729, which form the $HCO_3^-$ binding site, after which the ion could be released to the outside environment. Likewise, the processes of releasing $HCO_3^-$ and uptake of $Cl^-$ could be obligatorily coupled, where charge-charge repulsion between $HCO_3^-$ and $Cl^-$ could facilitate freeing the $HCO_3^-$ ion from the OF state of band 3 (Fig 7).

Several possible mechanisms have been proposed for the band 3 transporter to shuttle ions across the membrane [34–37]. Our proposed mechanism, based on the cryo-EM structural data, is in good agreement with the predicted elevator-type mechanism [33,38,39], where a switch of conformation from the OF to IF states and vice versa gives rise to a vertical change in position of bound ions with respect to the membrane surface. This elevator mechanism is also consistent with known experimental data of the red cell band 3 exchanger.

## Materials and methods

### Human erythrocyte membrane protein extraction

Human erythrocytes (RBCs) were purchased from ZenBio (Durham, North Carolina, United States of America). Freshly packed cells (5 ml) were washed twice with phosphate-buffered saline (PBS) buffer and then hypotonically lysed with 5 mM potassium phosphate buffer (pH 7.4). The membranes were separated by centrifugation at $100,000 \times g$ for 30 min, washed twice with 20 mM HEPES-NaOH (pH 7.5) followed by centrifugation at $100,000 \times g$ rpm for 30 min. The membrane protein was then solubilized in 1% (w/v) n-Dodecyl-β-D-Maltoside (DDM) and 0.1% cholesteryl hemisuccinate (CHS) for 1 h at 4°C. Insoluble material was removed by ultracentrifugation at $100,000 \times g$. The extracted cell membrane that contained a variety of unknown membrane proteins was concentrated to 10 mg/ml in a buffer containing 20 mM Tris-HCl (pH 7.5), 100 mM NaCl, 0.05% DDM, and 0.005% CHS.

### Nanodisc preparation

To reconstitute erythrocyte membrane proteins into nanodiscs, the membrane proteins were mixed with the scaffold protein MSP1E3D1 and POPC:POPE:POPG (3:1:1) lipid mix at a molar ratio of 1:2:70 for 15 min at room temperature, and 0.8 mg/ml prewashed bio-beads (Bio-Rad) was subsequently added, and the mixture was incubated for 1 h on ice followed by

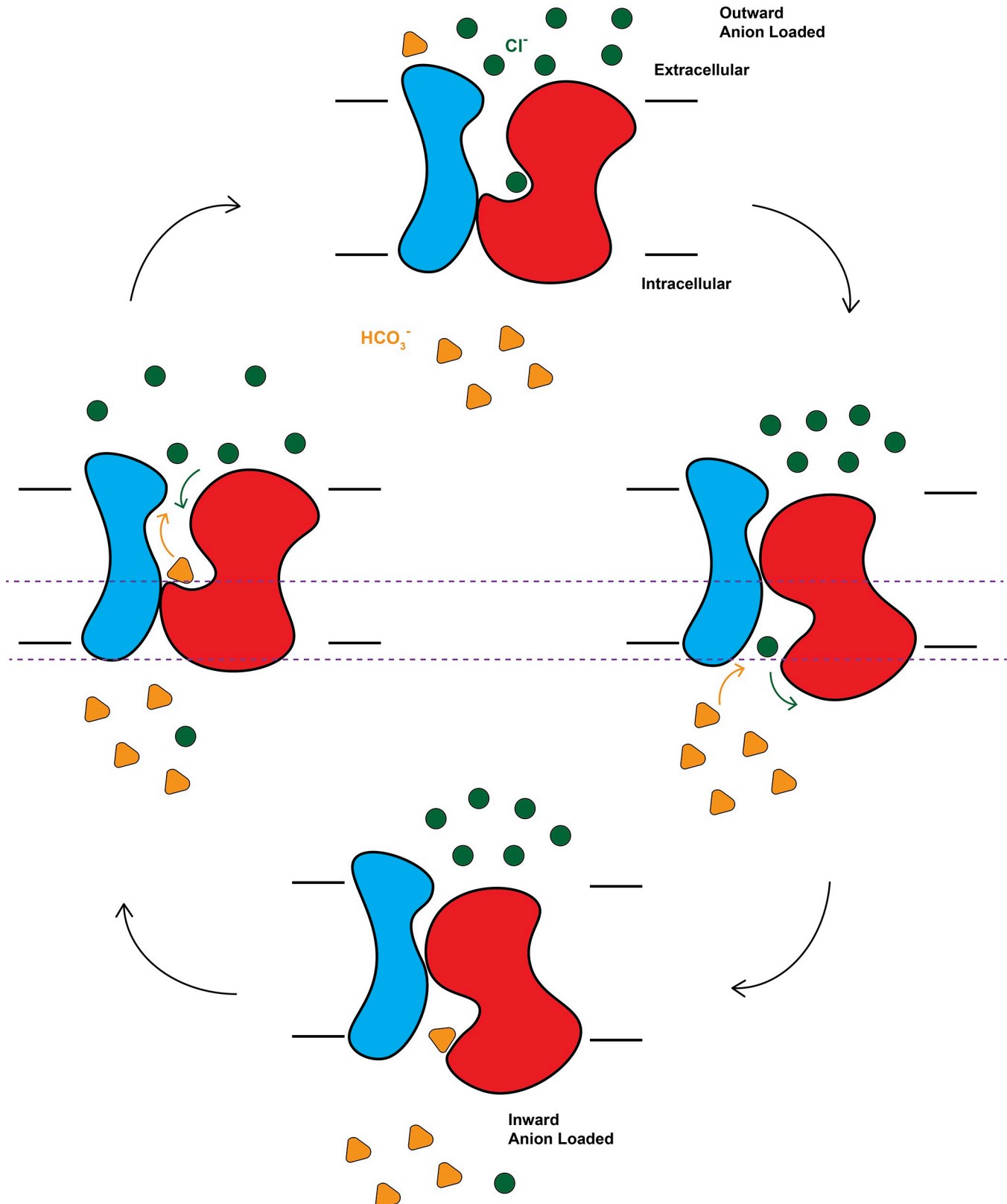

**Fig 7. Proposed mechanism for Cl⁻ and HCO₃⁻ translocations via band 3.** This schematic diagram indicates that the first step for Cl⁻ transport may involve the accessibility of Cl⁻ into the transporter via the OF state. Binding of Cl⁻ may trigger a conformational change to switch band 3 to the IF state via a rigid-body

rocking movement of the transport domain with respect to the channel domain of band 3. This movement leads to a downward shift of bound Cl$^-$ by approximately 5.5 Å toward the inside of the RBC. The bound Cl$^-$ ion could then be released into the cell. The deliverance of Cl$^-$ into and collection of HCO$_3^-$ from the RBC via the IF state of band 3 could be coupled with each other, because charge-charge repulsion between these 2 ions could help expel Cl$^-$ into the cell. After releasing Cl$^-$ and picking up HCO$_3^-$, band 3 may then shift its conformation back to the OF state. This movement is also caused by a rigid-body rocking motion of the transport domain in relation to the channel domain, resulting in an upward shift of the location of R730 by approximately 14 Å toward the outside of the RBC. After, the ion could be released to the outside environment. Likewise, the processes of releasing HCO$_3^-$ and uptake of Cl$^-$ could be obligatorily coupled, where charge-charge repulsion between HCO$_3^-$ and Cl$^-$ could facilitate freeing the HCO$_3^-$ ion from the OF state of band 3. IF, inward-facing; OF, outward-facing; RBC, red blood cell.

overnight incubation at 4°C. The protein-nanodisc solution was filtered through 0.22 μm nitrocellulose filter tubes to remove the biobeads. A superose 6 column (GE Healthcare) equilibrated with 20 mM Tris-HCl (pH 7.5) and 100 mM NaCl was then used to separate the free nanodiscs from the protein-nanodisc complexes. For the band 3-bicarbonate complex, the protein solution was load into a superose 6 column equilibrated with 20 mM HEPES-NaOH (pH 7.5) and 100 mM NaHCO$_3$. Peak fractions were pooled, concentrated to 1 mg/ml and immediately used to prepare grids for cryo-EM data collection.

The RBC membrane samples were applied to glow-discharged holey carbon grids (Quantifoil R1.2/1.3 Cu 300 mesh), blotted for 5 to 15 s, and then plunge-frozen in liquid ethane using a Vitroblot (Thermo Fisher). The grids were transferred into cartridges, and images were recorded at 1 to 2.0 μm underfocus on a K3 direct electron detector (Gatan) with super-resolution and correlated double sampling (CDS) mode at nominal 81 K magnification corresponding to a sampling interval of 1.08 Å/pixel (super resolution 0.54 Å/pixel). Each micrograph was exposed for 4.0 s with 11.71 e$^-$/Å$^2$/sec dose rate (total specimen dose, 38 e$^-$/Å$^2$), and 40 frames were captured per specimen area using serialEM [40].

## Data processing

The RBC membrane were processed using an improved BaR protocol (S1 Fig). Super-resolution image stacks were aligned and binned using patch motion correction from the cryoSPARC program [41] with a binning factor of 2, to give a final pixel size of 1.08 Å /pixel. Contrast transfer functions (CTFs) were estimated using patch CTF in cryoSPARC [41]. After manual inspection to discard poor images and to estimate particle size, the blob picker in cryoSPARC [41] was used to select particles from subsets of micrographs. These particles were classified in 1 round of 2D classification, and clear templates were selected for template picking in cryoSPARC [41]. Template picker was used to select initial particle sets. Several iterative rounds of 2D classification were used to clean these particle sets with different circular masks to account for different particle sizes. Featureless classes were removed from each step to obtain cleaned heterogeneous particle stacks for further processing. The 2D class averages containing clear 2D features were selected and used to generate a Topaz [42] training model. The trained model was then used to pick particles using Topaz [42] extract in cryoSPARC [41].

From heterogeneous particle sets, particles were classified, and final maps were solved using the build and retrieve (BaR) [25] iterative methodology. In brief, ab initio methods were used to build initial 3D maps from the selected classified particles, and particles were then retrieved based on the maps. To build the initial maps, particles were separated using 2D classification paired with 3D ab initio and heterogeneous classifications. To increase particle counts, these initial classes were used to retrieve particles from the topaz-picked particle stack. 3D heterogeneous refinement using the ab initio classes, determined from the build phase of BaR, was applied to the cleaned heterogeneous particle sets. The new particle subsets were then cleaned using multiple rounds of 2D and 3D ab initio classifications. Nonuniform refinement using cryoSPARC [41] was used to refine 3D reconstructions. A soft mask that covers

the band 3 dimer was used for local refinement. The resulting maps corresponding to the major membrane protein exhibited high quality, allowing for the construction of models and identification of the protein using Model Angelo [43].

## Molecular dynamics (MD) simulations

The protonation states of the titratable residues of the band 3 exchanger were determined using the H++ server (http://biophysics.cs.vt.edu/). The OF monomer of the OF-IF Cl⁻ bound band 3 structure was immersed in an explicit lipid bilayer consisting of POPC and POPE with a molecular ratio of 1:1 and a water box with dimensions of 111.6 Å x 109.8 Å x 114.3 Å using the CHARMM-GUI Membrane Builder webserver (http://www.charmm-gui.org/?doc=input/membrane) [44]. We then added 300 mM NaCl and extra neutralizing counter ions into the system for simulations. The total number of atoms were 104,882. The tleap program was used to generate parameter and coordinate files using the ff14SB and Lipid17 force fields for the protein and lipids, respectively. The PMEMD.CUDA program implemented in AMBER18 (AMBER 2018, UCSF) was used to conduct MD simulations. The simulations were performed with periodic boundary conditions to produce isothermal-isobaric ensembles. Long-range electrostatics was calculated using the particle mesh Ewald (PME) method [45] with a 10 Å cutoff. Prior to the calculations, energy minimization of the system was carried out. Subsequently, the system was heated from 0 K to 303 K using Langevin dynamics with the collision frequency of 1 ps⁻¹. During heating, the OF state band 3 monomer was position-restrained using an initial constant force of 500 kcal/mol/Å² and weakened to 10 kcal/mol/Å² in order to allow for the movement of lipid and water molecules. The system was then going through 5 ns equilibrium MD simulations. Finally, a total of 1 µs production MD simulations were conducted. During simulations, the coordinates were saved every 200 ps for analysis. The system was well equilibrated after 200 ns simulations according to root mean square deviations (RMSDs) of the exchanger's Cα atoms. GROMCAS analysis tools were used for the MD simulation trajectory analysis [46]. The snapshot of the last frame of the MD simulation trajectory was extracted for Cl⁻ and band 3 interaction energy calculations using the Discovery Studio 2017 program (Biovia Corp., San Diego, California, USA). The Cl⁻ bound OF band 3 monomer structure was minimized using the generalized Born with implicit membrane (GBIM) of the CHARMM force field for 200 steps with smart minimizer optimization and followed by interaction energy calculations. The binding site residues were defined by within 6.5 Å of the bound Cl⁻ ion. These interaction energies mainly account for van der Waals and electrostatic interactions between bound Cl⁻ and binding residues (S2 Table). The electrostatic interactions were calculated using a distance-dependent dielectric constant ($\varepsilon = 2r$) as an implicit solvent model.

## Supporting information

**S1 Table. Cryo-EM data collection, processing, and refinement statistics.**
(PDF)

**S2 Table. Interaction energies within 6.5 Å of bound Cl⁻.**
(PDF)

**S1 Fig. Band 3 in the presence of 100 mM Cl⁻ processing workflow.** (A) Representative 2D classes of band 3. (B) Processing of 17,814 micrographs using the BaR protocol allowed us to get initial pool of 10,862,923 particles. Further 2D classification led to the selection of 926,480 particles. Nonuniform refinement, 3D focused classification, and focused refinement resulted

in the high-resolution structures of band 3 in the OF-IF and IF-IF conformational states.
(TIF)

**S2 Fig. Overall structure and topology of each subunit of band 3.** (A) Secondary structural elements of a subunit of band 3. An N-linked glycosylation site is found at residue N642 of each subunit of band 3. (B) Topology of a subunit of band 3. The transmembrane domain of the band 3 subunit contains 14 TMs. The N-linked modification site at residue N642 is highlighted with a green hexagon. In both (A) and (B), TMs 1–7 (orange) and TMs 8–14 (purple) are structurally related to each other in that they are arranged into 2 inverted repeats.
(TIF)

**S3 Fig. GS-FSC resolution and local cryo-EM maps of the OF-IF structure of band 3 in the presence of Cl⁻.** (A) GS-FSC resolution of the cryo-EM map. (B) Euler angle distribution. (C) Local cryo-EM density map of the OF-IF structure of band 3. (D) Cryo-EM density maps of bound Cl⁻, bound lipids (PC and PIP2), and NAG at the glycosylated modification site.
(TIF)

**S4 Fig. Comparison of the structures of human band 3 and bovine band 3.** (A) Superimposition of the OF subunit of human band 3 (red) to that of bovine band 3 (green) gives rise to an r.m.s.d. of 1.07 Å (for 422 Cα atoms). (B) Superimposition of the IF subunit of human band 3 (orange) to that of bovine band 3 (cyan) gives rise to an r.m.s.d. of 1.28 Å (for 422 Cα atoms).
(TIF)

**S5 Fig. Results of the 1 μs MD simulations.** The simulations show results of the OF monomer of band 3 bound with Cl⁻. The Cα atoms RMSD (root mean square deviation) are based on the MD simulation trajectories (1 μs). The simulations have been done for only one time.
(TIF)

**S6 Fig. The Cl⁻ binding site.** MD simulations suggest that the bound Cl⁻ ion at the Cl⁻ binding site of band 3 is coordinated with 6 water molecules. Important residues responsible for binding Cl⁻ are in cyan sticks.
(TIF)

**S7 Fig. GS-FSC resolution and local cryo-EM maps of the IF-IF structure of band 3 in the presence of Cl⁻.** (A) GS-FSC resolution of the cryo-EM map. (B) Euler angle distribution. (C) Local cryo-EM density map of the OF-IF structure of band 3. (D) Cryo-EM density maps of bound lipids (PC and PIP2) and NAG at the glycosylated modification site.
(TIF)

**S8 Fig. Band 3 in the presence of 100 mM HCO₃⁻ processing workflow.** (A) Representative 2D classes of band 3. (B) Processing of 43,385 micrographs using the BaR protocol allowed us to get initial pool of 48,595,748 particles. Further 2D classification led to the selection of 203,517 particles. Nonuniform refinement, 3D focused classification, and focused refinement resulted in the high-resolution structures of band 3 in the OF-IF, IF-IF, and OF-OF conformational states.
(TIF)

**S9 Fig. GS-FSC resolution and local cryo-EM maps of the IF-IF structure of band 3 in the presence of HCO₃⁻.** (A) GS-FSC resolution of the cryo-EM map. (B) Euler angle distribution. (C) Local cryo-EM density map of the IF-IF structure of band 3. (D) Cryo-EM density maps of bound lipids (PC and PIP2) and NAG at the glycosylated modification site.
(TIF)

**S10 Fig. GS-FSC resolution and local cryo-EM maps of the OF-IF structure of band 3 in the presence of HCO$_3^-$.** (A) GS-FSC resolution of the cryo-EM map. (B) Euler angle distribution. (C) Local cryo-EM density map of the OF-IF structure of band 3. (D) Cryo-EM density maps of bound HCO$_3^-$, bound lipids (PC and PIP2), and NAG at the glycosylated modification site.
(TIF)

**S11 Fig. GS-FSC resolution and local cryo-EM maps of the OF-OF structure of band 3 in the presence of HCO$_3^-$.** (A) GS-FSC resolution of the cryo-EM map. (B) Euler angle distribution. (C) Local cryo-EM density map of the OF-OF structure of band 3. (D) Cryo-EM density maps of bound HCO$_3^-$, bound lipids (PC and PIP2), and NAG at the glycosylated modification site.
(TIF)

**S1 Movie. Shift in conformations between the outward-facing and inward-facing states.** The major conformational change between these 2 states can be attributed to the rigid-body rocking motion of the core domain (orange) in relation to the gate domain (blue).
(MP4)

## Acknowledgments

We are grateful to the Cryo-Electron Microscopy Core at the CWRU School of Medicine and Dr. Kunpeng Li for access to the sample preparation and Cryo-EM instrumentation. The computations were supported by the ITS (Information Technology Services) Research Computing at Northeastern University and the Argonne Leadership Computing Facility (ALCF) at Argonne National Laboratory. We thank Dr. Philip Klenotic for proofreading the manuscript.

## Author Contributions

**Conceptualization:** Chih-Chia Su, Edward W. Yu.

**Data curation:** Chih-Chia Su, Zhemin Zhang, Meinan Lyu, Meng Cui.

**Formal analysis:** Chih-Chia Su, Zhemin Zhang, Meinan Lyu, Meng Cui, Edward W. Yu.

**Funding acquisition:** Edward W. Yu.

**Investigation:** Chih-Chia Su, Zhemin Zhang, Meinan Lyu, Meng Cui, Edward W. Yu.

**Project administration:** Edward W. Yu.

**Supervision:** Edward W. Yu.

**Validation:** Chih-Chia Su, Zhemin Zhang, Meinan Lyu, Meng Cui, Edward W. Yu.

**Visualization:** Chih-Chia Su, Zhemin Zhang, Meinan Lyu, Meng Cui, Edward W. Yu.

**Writing – original draft:** Edward W. Yu.

**Writing – review & editing:** Chih-Chia Su, Zhemin Zhang, Meinan Lyu, Edward W. Yu.

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
