## [Editor Report · Decision Letter 0]

1 Sep 2023

Dear Dr Yu, 

Thank you for submitting your manuscript entitled "Transport mechanism of band 3 involves the concerted movement of chloride and bicarbonate" for consideration as a Research Article by PLOS Biology. Please accept my apologies for the delay in getting back to you as we consulted with an academic editor about your submission.

Your manuscript has now been evaluated by the PLOS Biology editorial staff, as well as by an academic editor with relevant expertise, and I am writing to let you know that we would like to send your submission out for external peer review.

Once your full submission is complete, your paper will undergo a series of checks in preparation for peer review. After your manuscript has passed the checks it will be sent out for review. To provide the metadata for your submission, please Login to Editorial Manager (https://www.editorialmanager.com/pbiology) within two working days, i.e. by Sep 03 2023 11:59PM.

Kind regards,

Richard

Richard Hodge, PhD

rhodge@plos.org

PLOS

---

## [Decision Letter · Decision Letter 1]

26 Oct 2023

Dear Dr Yu,

Thank you for your patience while your manuscript "Transport mechanism of band 3 involves the concerted movement of chloride and bicarbonate" was peer-reviewed at PLOS Biology. Please accept my apologies for the delays that you have experienced during the peer review process. Your manuscript has now been evaluated by the PLOS Biology editors, an Academic Editor with relevant expertise, and by four independent reviewers. 

In light of the reviews, which you will find at the end of this email, we would like to invite you to revise the work to thoroughly address the reviewers' reports.

As you will see, the reviewers think the findings are interesting and note that the structures are well-done and provide significant insights into the mechanisms of anion recognition and exchange. However, Reviewer #2 raises concerns with the fine interpretation of the results and the assignment of the ligand cofactors in the structural model, since it is not clear how deprotonated glutamate residues could contribute to chloride ion stabilization. He/she suggests that additional experiments, such as MD simulations or functional transport assays, are included to address this. After discussions with the Academic Editor, we will not make the inclusion of MD simulations essential for publication. However, we feel that functional experiments would provide useful and valuable information and we ask that these assays are included in the revised version. 

Given the extent of revision needed, we cannot make a decision about publication until we have seen the revised manuscript and your response to the reviewers' comments. Your revised manuscript is likely to be sent for further evaluation by all or a subset of the reviewers.

**IMPORTANT - SUBMITTING YOUR REVISION**

*Re-submission Checklist*

*Published Peer Review*

*PLOS Data Policy*

*Blot and Gel Data Policy*

Sincerely,

Richard

Richard Hodge, PhD

rhodge@plos.org

REVIEWS:

Reviewer #1 (Motoyuki Hattori, signs review): Su et al. reported the cryo-EM structures of the band 3 transporter in multiple conformations, including a newly determined inward conformation. It is noteworthy that they used a self-developed technique called BaR. The quality and significance of the work would merit publication in PLOS biology. Therefore, I recommend this paper for publication if the following concerns are adequately addressed. 

Main concerns

1. TM helix numbering labels

Page 6 " TMs 1-7 and TMs 8-14 are structurally related to each other in that they are arranged into two inverted repeats.". 

As shown above as an example, the authors have extensively mentioned the numbering of the TM helices, but most of the figures lack this information, so it was often difficult to understand. I strongly recommend the authors to add another figure to show the monomeric structure with the labels for each TM helix together with the membrane topology cartoon. In addition, I would add more labels for the TM helix numbering in Figures 2-6.

2. Movies to illustrate the structural changes in the transport cycle

The authors should prepare a movie illustrating the structural changes in the transport cycle using their structures.

3. Page 12. " Surprisingly, we only observed these bound ions in the outward-facing conformation, and

not in the inward-facing state, of the band 3 protomers."

Please explain in detail why the authors did not see bound ions in the inward-facing state.

Reviewer #2: In this contribution by Su et al., the collection of five cryo-EM structures of human band 3 transporter is reported. Although I have no concerns about the quality of generated models, and appreciate the fact that the authors managed to catch this transporter in different conformational states, which certainly improves our understanding of the transport mechanism, I do have some concerns regarding the fine interpretation of their results. This is especially valid for the assignment of cofactors, which I am very doubtful about. I fail to grasp how the negatively charged side chain of glutamate E681 can be a main binder of a negatively charged chloride ion. The latter should rather repel than being attracted to E681. The same problem is with the proposed role of E535 as a recruiter of a chloride ion. There is a general problem with cryo-EM studies that there is no direct way to confirm the identity of a ligand, especially when they are small in size and at relatively low resolution, in contrast to e.g. crystallography, where anomalous dispersion can be measured to confirm the identity. In this particular work it is impossible to unambiguously assign the ligands (especially chloride, to a lesser extent bicarbonate), so the authors should either conduct additional experiments, e.g., MD simulations, transport assays, etc, or they should significantly strip down the discussion of ligand assignment. 

Minor concerns: please avoid statements of high resolution, 3 Å is anyway not high (even for EM these days)

In the methods: a typo in the word 'pooled' 

Fig S2/S3/S5 panel C , the colour of the protein or density should be changed 

Reviewer #3: The manuscript of Su and Yu et al titled "Transport mechanism of band 3 involves the concerted movement of chloride and bicarbonate" reports cryoEM structures of the band 3 protein extracted from human red blood cells and reconstituted into nanodiscs. The purification process seems to produce predominantly homodimeric the human band 3 protein that does not contain any associated proteins. The authors developed and applied a procedure that they call the BaR methodology to tease out multiple conformations within the same data set. In the presence of chloride ion, two conformations were resolved, termed OF-OF and OF-IF, while in the presence of bicarbonate ion, three conformations were resolved, termed OF-OF, IF-IF, and OF-IF. The authors then defined a binding site for chloride and bicarbonate ions, respectively, which relies mainly on the presence or absence of a density at the "internal cavity" where TM3 and TM10 meet. Chloride and bicarbonate ions seem to occupy discrete sites within the "internal cavity" region, and the position of the bicarbonate ion may shift slightly depending on whether the structure is in IF or OF. 

Overall, the work on structure determination is of good quality and the analysis and interpretation of the structures are cautious and reasonable. The work is highly significant for its novel information on mechanisms of anion recognition and exchange, and for the methods developed on isolation of native protein and sorting of particles and building of structural models. Although I would love to see targeted functional validations of novel conclusions on discrete positions of the two anions and the movement of the domains, I understand that these studies are challenging and would take time, and that these could be completed in the future. 

I have a few minor comments for the authors to consider:

The name "gate" and "core" domains were given in the initial UraA paper when the mechanism of transport was vague. I think the field of UraA and its structurally homologous relatives has settled on "interface" and "transport" domains. 

The "internal cavity" between TM3 and TM10 is often referred to as the "crossover" region. 

The Abstract is misleading by stating "To elucidate …. in its native form, human ghost membrane was applied to a cryo-EM holey carbon grid to define its composition". While it is true that the band 3 protein is from a native source, it is no longer in its native form after detergent extraction and nor was the ghost membrane directly applied to the grid. When I first read the Abstract, I was expecting that authors would report associated proteins to band 3 or tightly bound native lipids and so on. 

The word "concerted" is prone to misunderstanding. I understand that the authors use the word to describe a process in which a bound anion is knocked off by an incoming anion, but I think "obligatorily coupled" might work.

Reviewer #4: In recent years there have been a flurry of papers describing the structure of the dimeric human Band 3 membrane protein using single particle cryo-electron microscopy. Two of these papers [Refs. 22 & 39] describe the ankyrin complex that includes the structure of the entire Band 3 protein interacting with Protein 4.2 and the membrane cytoskeleton (Vallese et al. (2022) https://pubmed.ncbi.nlm.nih.gov/35835865/; Xia et al. (2022) https://www.nature.com/articles/s41594-022-00779-7). An important consideration in these studies is that the cytoplasmic domain of purified Band 3 is unstructured in the absence of an interaction with Band 4.2. Another paper [Ref. 18] describes the structure of the membrane domain of Band 3 with a series of bound inhibitors (Capper, et al. (2023) https://pubmed.ncbi.nlm.nih.gov/37679563/). An additional paper [Ref 23] describes the structure of the membrane domain bovine Band 3 at resolutions of 4.5-7.1 Å in support of an elevator mode of transport (Zhekova et al. (2022) https://www.nature.com/articles/s42003-022-04306-8). This paper describes Band 3 in the inward facing conformation in addition to the outward facing conformation. The paper also describes a Band 3 dimer in which the subunits are in the outward and inward-facing conformation. The first description of the crystal structure of the isolated membrane domain of human Band 3 in the outward-facing state with bound inhibitor was published in 2015 (Arakawa et al. (2015) https://www.science.org/doi/10.1126/science.aaa4335). 

So, what is new in this paper?

1. The protein is extracted from membranes with detergent and reconstituted into nanodiscs that retain endogenous lipids that were revealed in the structure, included PIP2 molecules at the dimer interface as predicted from MD simulation studies.

2. The paper provides the structure of the human Band 3 in five different conformational states at a resolution of Å, including an asymmetrical dimer in both the outward (OF) and inward-facing (IF) states. 

3. The OF/IF structure (#1) at 2.99 Å resolution revealed the positions of chloride close to Asp681 in the outward-facing state while the inward-facing state was devoid of substrates.

4. The IF/IF state (#2) at 2.97Å resolution was also devoid of substrates, corresponding to the empty inward-facing state. 

5. IF/IF (#3), OF/IF (#4) and OF/OF (#5) structures were obtained in the presence of bicarbonate. IF/IF (#3) at 2.99 Å resolution was also empty like #2. The OF/IF (#4) state at 3.12 Å resolution had bound bicarbonate interacting with Arg730 as expected, while the IF subunit was once again empty. The OF/OF (#5) state at 3.16 Å resolution contained bicarbonate in both subunits. 

6. The author note a change in the conformation of TM10 assuming a random state in the inward-facing conformation moving Arg730 ~14 Å towards the inside of the cell.

7. The authors conclude that the structures support an elevator mode of transport. 

This paper provides five highest resolution structures of human Band 3 in various conformational states, including an asymmetrically OF/IF form obtained to date. The empty and OF state with bound chloride and bicarbonate is provided as is the empty IF state. The binding sites for chloride and bicarbonate are described in exquisite molecular detail. It may be difficult to capture the IF state with bound chloride or bicarbonate due to the lower affinity for these ions and accessibility to solvent. 

Minor Comments

1. I would include the word "human" in the title and in the abstract as appropriate. 

2. Only the membrane domain was resolved in these structures and this should be made clear in the abstract and early in the paper.

3. I would recommend adding a reference in the Introduction to the excellent review "Cell physiology and molecular mechanism of anion transport by erythrocyte band3/AE1" by Michael Jennings (https://journals.physiology.org/d

---

## [Editor Report · Decision Letter 2]

11 May 2024

Dear Dr Yu,

Thank you for your patience while we considered your revised manuscript "Transport mechanism of human band 3 involves the obligatorily coupled movement of chloride and bicarbonate" for publication as a Research Article at PLOS Biology. This revised version of your manuscript has been evaluated by the PLOS Biology editors and the Academic Editor.

Based on our Academic Editor's assessment of your revision, I am pleased to say that we are likely to accept this manuscript for publication, provided you satisfactorily address the remaining comment raised by the Academic Editor regarding the MD simulations (pasted below my signature). Please also make sure to address the following data and other policy-related requests that I have provided below (A-F):

(A) We would like to suggest the following modification to the title:

"Cryo-EM structures of the human band 3 transporter indicate a transport mechanism involving the coupled movement of chloride and bicarbonate ions”

(B) You may be aware of the PLOS Data Policy, which requires that all data be made available without restriction: http://journals.plos.org/plosbiology/s/data-availability. For more information, please also see this editorial: http://dx.doi.org/10.1371/journal.pbio.1001797

-Supplementary files (e.g., excel). Please ensure that all data files are uploaded as 'Supporting Information' and are invariably referred to (in the manuscript, figure legends, and the Description field when uploading your files) using the following format verbatim: S1 Data, S2 Data, etc. Multiple panels of a single or even several figures can be included as multiple sheets in one excel file that is saved using exactly the following convention: S1_Data.xlsx (using an underscore).

-Deposition in a publicly available repository. Please also provide the accession code or a reviewer link so that we may view your data before publication. 

Figure S5

(C) Thank you for providing the structural data in the PDB and EMDB database. However, we note that the structures are currently on hold for release. We ask that you please make the structures publicly available at this stage before publication.

(D) Please also ensure that each of the relevant figure legends in your manuscript include information on *WHERE THE UNDERLYING DATA CAN BE FOUND*, and ensure your supplemental data file/s has a legend.

(E) Per journal policy, if you have generated any custom code during the curse of this investigation, please make it available without restrictions upon publication. Please ensure that the code is sufficiently well documented and reusable, and that your Data Statement in the Editorial Manager submission system accurately describes where your code can be found.

(F) Please ensure that your Data Statement in the submission system accurately describes where your data can be found and is in final format, as it will be published as written there. 

We expect to receive your revised manuscript within two weeks. 

*Published Peer Review History*

*Press*

Kind regards,

Richard

Richard Hodge, PhD

rhodge@plos.org

COMMENTS FROM THE ACADEMIC EDITOR

The only remark I have concerns the stated binding affinity of -20 kcal/mol obtained from MD simulations, which would correspond to a Kd of 2 10-15 M which is not realistic. The authors should thus not take this value quantitatively but emphasize that it reflects the general properties of the site to coordinate an anion.

---

## [Editor Report · Decision Letter 3]

20 Jun 2024

Dear Dr Yu,

Thank you for the submission of your revised Research Article "Cryo-EM structures of the human band 3 transporter indicate a transport mechanism involving the coupled movement of chloride and bicarbonate ions" for publication in PLOS Biology. On behalf of my colleagues and the Academic Editor, Raimund Dutzler, I am pleased to say that we can in principle accept your manuscript for publication, provided you address any remaining formatting and reporting issues. These will be detailed in an email you should receive within 2-3 business days from our colleagues in the journal operations team; no action is required from you until then. Please note that we will not be able to formally accept your manuscript and schedule it for publication until you have completed any requested changes.

PRESS

Sincerely, 

Suzanne

Suzanne De Bruijn, PhD, PhD

Senior Editor

PLOS Biology

rhodge@plos.org